# Bending and breaking of stripes in a charge ordered manganite

Benjamin H. Savitzky[1], Ismail El Baggari[1], Alemayehu S. Admasu[2], Jaewook Kim[2], Sang-Wook Cheong[2], Robert Hovden[3,5] & Lena F. Kourkoutis[3,4]

In charge-ordered phases, broken translational symmetry emerges from couplings between charge, spin, lattice, or orbital degrees of freedom, giving rise to remarkable phenomena such as colossal magnetoresistance and metal–insulator transitions. The role of the lattice in charge-ordered states remains particularly enigmatic, soliciting characterization of the microscopic lattice behavior. Here we directly map picometer scale periodic lattice displacements at individual atomic columns in the room temperature charge-ordered manganite $Bi_{0.35}Sr_{0.18}Ca_{0.47}MnO_3$ using aberration-corrected scanning transmission electron microscopy. We measure transverse, displacive lattice modulations of the cations, distinct from existing manganite charge-order models. We reveal locally unidirectional striped domains as small as ~5 nm, despite apparent bidirectionality over larger length scales. Further, we observe a direct link between disorder in one lattice modulation, in the form of dislocations and shear deformations, and nascent order in the perpendicular modulation. By examining the defects and symmetries of periodic lattice displacements near the charge ordering phase transition, we directly visualize the local competition underpinning spatial heterogeneity in a complex oxide.

[1] Department of Physics, Cornell University, Ithaca, NY 14853, USA. [2] Rutgers Center for Emergent Materials and Department of Physics and Astronomy, Rutgers University, Piscataway, NJ 08854, USA. [3] School of Applied and Engineering Physics, Cornell University, Ithaca, NY 14853, USA. [4] Kavli Institute for Nanoscale Science, Cornell University, Ithaca, NY 14853, USA. [5] Present address: Department of Materials Science & Engineering, University of Michigan, Ann Arbor, MI 48109, USA. Benjamin H. Savitzky and Ismail El Baggari contributed equally to this work. Correspondence and requests for materials should be addressed to L.F.K. (email: lena.f.kourkoutis@cornell.edu)

Charge density wave (CDW) states are periodic modulations of both the electron density and atomic lattice positions. These states epitomize emergent order via electron–lattice interaction, and have taken a central role in understanding exotic phenomena in complex materials. CDWs mediate metal–insulator transitions, compete with high-temperature superconductivity, and underlie the mechanism of colossal magnetoresistance in manganites[1–6]. Mounting evidence indicates that nanoscale spatial inhomogeneity between competing electronic phases plays a fundamental role in complex electronic systems quite broadly[7–9]. For example, local competition and coexistence between charge-ordered and ferromagnetic regions is responsible for the colossal magnetoresistance effect in manganites, while in cuprates, the suppression of superconducting order coincides with the emergence of charge-ordered patches[2,10,11]. However, understanding of the microscopic mechanism driving such competition is lacking, requiring local interrogation of the atomic-scale behavior.

The manganese oxides provide a practical test bed for universal CDW phenomenology, as their strong electron–lattice coupling results in relatively robust charge and spin-ordered phases[12]. Striped states have been imaged in manganites with dark-field transmission electron microscopy (DF-TEM), however, resolution and signal-to-noise are limited in DF-TEM because electrons are collected from a small window of momentum space[13,14]. Moreover, the contrast mechanism of DF-TEM complicates interpretation, yielding inconsistent models of the modulation structure, including organization of $Mn^{3+}$ and $Mn^{4+}$ sites, continuous charge density modulations which pin to lattice defects, and $Mn^{3+}$ pairs coupled by an adjoining hole[13–16]. Atomically resolved measurements of periodic lattice displacements (PLDs) modulating the atomic lattice positions are therefore required.

Here we quantitatively map picometer scale (<10 pm) PLDs at individual atomic columns in the charge-ordered manganite $Bi_{0.35}Sr_{0.18}Ca_{0.47}MnO_3$ (BSCMO) near its transition temperature using scanning transmission electron microscopy (STEM). In contrast to proposed manganite charge-order models[13,16–19], our data show displacive, transverse, periodic modulations of the cation sites, with amplitudes of 6.2 pm/8.2 pm on the A/B sites of the perovskite lattice. We find two coexisting PLDs, forming locally unidirectional domains as small as ~5 nm despite appearing bidirectional over larger length scales, a distinction which is important but often challenging to establish[10,14,20,21]. We unearth shear deformations and topological singularities in one PLD field, and establish that they coincide with nascent order in the perpendicular modulation. Our results directly visualize the nanoscale complexity arising from competing phases and provide insight into the microscopic nature of charge ordering[2,7–10,20].

## Results

### Experimental hallmarks of the charge-ordered state.
The BSCMO orthorhombic perovskite lattice (space group *Pnma*, Fig. 1a) is imaged in projection along the *b*-axis with aberration-corrected high-angle annular dark-field (HAADF)-STEM (Fig. 2a), which is sensitive to the Coulomb potential of the atomic nuclei; heavier Bi/Sr/Ca atomic columns (A-sites) appear brighter than lighter Mn columns (B sites) in the Z-contrast image. Temperature-dependent resistivity and magnetic susceptibility measurements on the host BSCMO crystal reveal an anomaly associated with charge ordering at $T_c = 315$ K and 318 K, respectively (Supplementary Figs. 1 and 2). Transport curves measured at zero field are nearly identical to those measured under application of a 2 T magnetic field, comparable to that of the microscope objective at the position of the specimen (Supplementary Fig. 1). Reflective polarized optical microscopy reveals

approximately 100 μm twin domains (Supplementary Fig. 3); STEM and electron diffraction are performed within a single twin domain.

Electron diffraction (Fig. 1b) shows a constellation of satellite peaks indicating two transverse, displacive PLDs (Fig. 1c, d) offsetting the atomic lattice with displacements

$$\mathbf{\Delta}_i(\mathbf{r}) = \mathbf{A}_i \sin(\mathbf{q}_i \cdot \mathbf{r} + \phi_i), \quad i \in \{1, 2\} \qquad (1)$$

where $\mathbf{A}_i$, $\mathbf{q}_i$, and $\phi_i$ are the PLD amplitude vector, wavevector, and phase, respectively, and $|\mathbf{q}_i| \approx \frac{1}{3}$ reciprocal lattice units (Supplementary Figs. 4 and 5). Note that valence modulations have been found to be minimal here (Supplementary Fig. 6) and elsewhere[15], therefore the state giving rise to the observed satellite peaks and accompanying the resistivity anomaly is referred to empirically as the charge-ordered or CDW state, agnostic to a particular underlying model. Field-free electron diffraction, with the objective lens turned off, showed no discernible changes in the superlattice structure (Supplementary Fig. 7), consistent with the magnetic field-dependent resistivity measurements and suggesting the charge-ordered state is robust to the applied magnetic field. Diffraction shows coexistence of the two orthogonal PLDs within a 1 μm selected area. A STEM Fourier transform (Fig. 1b) shows coexistence within a 30 nm field of view. In order to further investigate the local PLD structure, we extract the displacement vectors associated with each of the two modulations at every atomic site to generate the PLD maps shown in Fig. 2.

### Local structure of periodic lattice displacements.
To calculate the PLD fields $\mathbf{\Delta}_i(\mathbf{r})$ shown in Fig. 2, we first fit all atomic positions in our STEM data with ~2 picometer precision, an approach which has recently emerged as a powerful, quantitative characterization tool[22–24]. However, in contrast to prior STEM atom tracking work, the key challenge in mapping PLDs is defining an appropriate reference lattice, which is complicated by the presence of local PLD phase variations and multiple interpenetrating modulations. Our approach generates a reference image in which the contribution of a single modulation has been selectively removed, by damping all of the relevant satellite peaks from the Fourier transform of the original image. Fitting and subtracting corresponding lattice positions from the image pair yields $\mathbf{\Delta}_i(\mathbf{r})$ quantitatively. Damping the $q_1$ satellite peaks (Fig. 1b, c, blue arrows) generates a map of $\mathbf{\Delta}_1(\mathbf{r})$ (Fig. 2b), while damping the $q_2$ satellite peaks (Fig. 1b, d, red arrows) maps $\mathbf{\Delta}_2(\mathbf{r})$ (Fig. 2c). Simulations indicate that our method accurately reconstructs the PLD structure everywhere except at lattice sites directly adjacent to atomically sharp discontinuities in the PLD field. Analytical and algorithmic details, simulations, and error analysis are found in Supplementary Note 1 and Supplementary Figs. S8–15.

The microscopic structure of charge-ordered phases in manganites remains contested[16–19]; here the $\mathbf{\Delta}_1(\mathbf{r})$ map in Fig. 2b furnishes real-space evidence for displacive lattice modulations of both the Bi/Sr/Ca sites and the Mn sites, with respective amplitudes of 6.2 and 8.2 pm on the maximal sites (see Supplementary Fig. 16). The displacements are transverse to the modulation wavevector and generate a tripled unit cell. The historically prevailing model conjectures the localization and ordering of $Mn^{3+}$–$Mn^{4+}$ ions, which in turn activates an alternating compression and expansion of oxygen octahedra (Jahn–Teller effect)[13]. Other works propose the formation of Mn pairs (Zener polarons) with minimal valence modulations[15,16]. Our data suggest a different model. The strong structural modulation shown in Fig. 2b is consistent with the softening of a phonon mode, and the pattern of displacements provides a

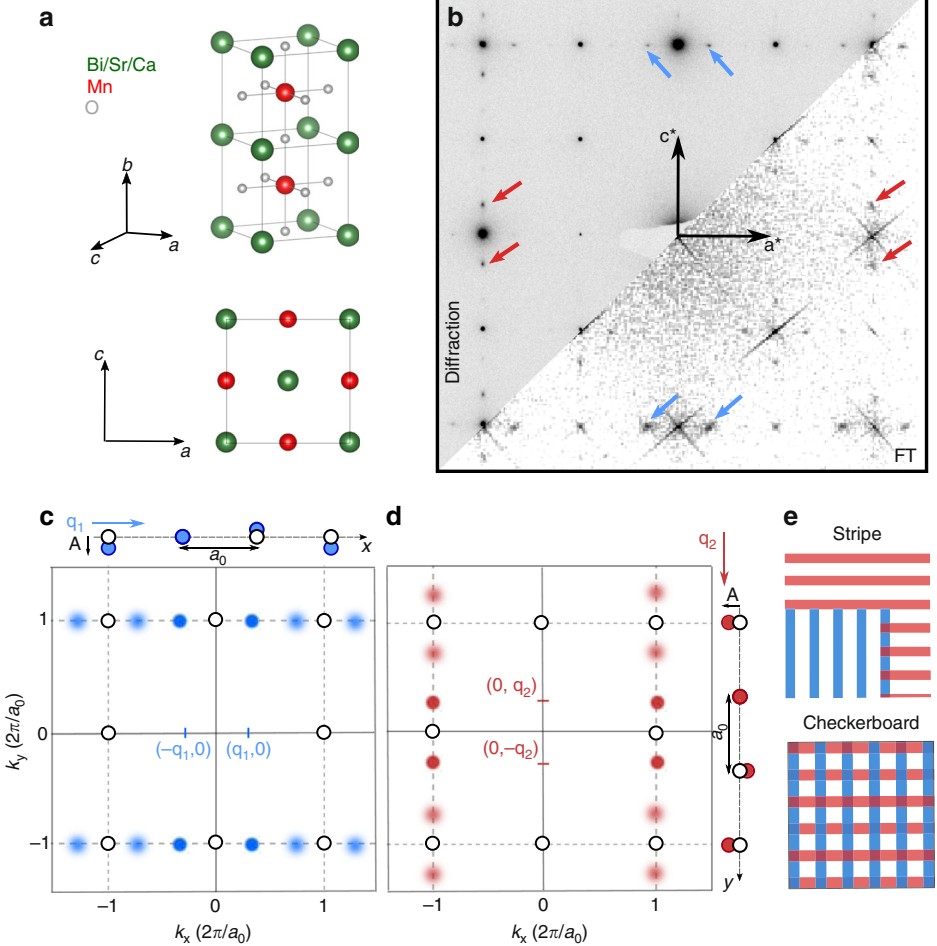

**Fig. 1** Periodic lattice displacements in reciprocal space. **a** The perovskite structure of $Bi_{0.35}Sr_{0.18}Ca_{0.47}MnO_3$ (BSCMO) and the projection of the unit cell along the $b$-axis. **b** Electron diffraction over a 1 µm selected area and the Fourier transform of a 30 nm field of view scanning transmission electron microscopy image of BSCMO along the $b$-axis. Satellite peaks corresponding to two transverse and displacive modulations with perpendicular wavevectors $q_1 \approx 1/3\ a^*$ and $q_2 \approx 1/3\ c^*$ are indicated by blue and red arrows, respectively. **c**, **d** Schematic of the Fourier transform of a square lattice (for simplicity) displaced by transverse modulations along $x$ and $y$, respectively. The intensity of a satellite peak is reduced when its reciprocal vector, $k = (k_x, k_y)$, is not parallel to the modulation polarization $A_i$ and vanishes when $k \cdot A_i = 0$. **e** Stripe states contain locally unidirectional modulations, while checkerboard states contain overlapping bidirectional modulations. Both stripe and checkerboard order are consistent with the reciprocal space data, which reflects the spatially averaged structure and cannot definitively determine the local symmetry

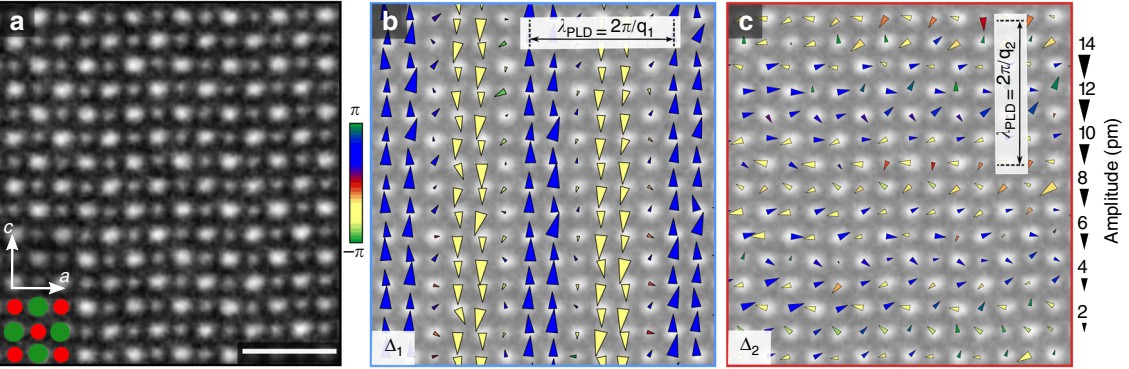

**Fig. 2** Mapping picometer scale, periodic displacements of atomic lattice sites. **a** High-angle annular dark-field scanning transmission electron microscopy projection image along the $b$-axis. The heavier (Bi, Sr, Ca) sites (green) appear brighter than the lighter Mn sites (red). **b** Mapping picometer scale periodic lattice displacements (PLDs) $\Delta_1(r)$ at each atomic lattice site in response to a single modulation wavevector $q_1$. PLD maps indicate a displacive modulation rather than an intensity modulation (cation order, charge disproportionation) with transverse polarization and $3a$ periodicity. Triangles represent displacements, with the area scaling linearly with displacement amplitude. The color represents the angle of the polarization vector, $A_1$, relative to the modulation wavevector, $q_1$, where blue (yellow) correspond to 90° (−90°) as indicated in the colorbar. **c** Map of $\Delta_2(r)$ displacements at each atomic lattice site in response to $q_2$ in the same region as **a**, **b**. The significantly weaker $\Delta_2(r)$ response is characteristic of locally striped, rather than checkerboard, ordering. The scale bar corresponds to 1 nm

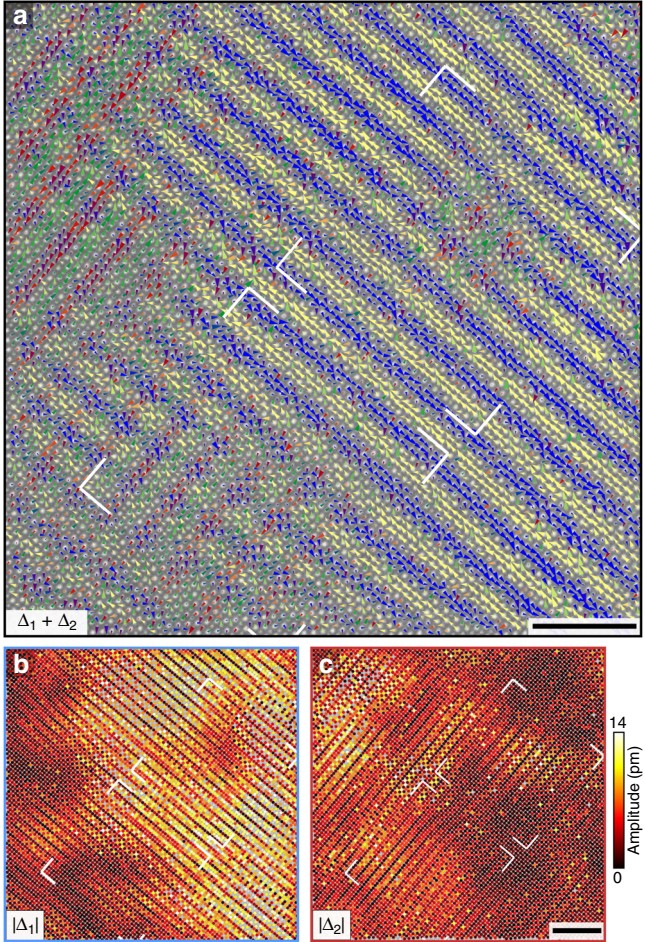

**Fig. 3** Nanoscale domain structure and local symmetry of periodic lattice displacement (PLD) stripes. **a** Combined PLD map showing the displacements $\Delta(\mathbf{r}) = \Delta_1(\mathbf{r}) + \Delta_2(\mathbf{r})$ at all ~9000 atomic sites in the 30 nm field of view. Colors indicates the displacement polarizations relative to $\mathbf{q}_1$ following the colorbar in Fig. 2, and triangle areas scale linearly with the displacement magnitudes. **b, c** Maps of the magnitudes $|\Delta_1(\mathbf{r})|$ and $|\Delta_2(\mathbf{r})|$ of the displacements due to each PLD individually reveals that the two PLD strengths are anti-correlated: when one is strong, the other is weak. The PLDs are stripe ordered, segregated into nanoscopic domains. The regions indicated by white delimiters contain local defect structures, which are further analyzed in Figs. 4 and 5. The scale bars correspond to 4 nm

structural model to further investigate the microscopic origin of the modulated state.

The superposition of multiple modulations can further mask the underlying microscopic mechanism behind PLD formation. For instance, distinguishing overlapping modulations (checkerboards) from spatially anti-correlated unidirectional domains (stripes) is essential but challenging, as both have the same spatially averaged symmetry (Fig. 1b–d)[14,20,21,25,26]. Our data clearly indicates that locally, BSCMO forms striped states: where one PLD is suppressed, the other is strong, starkly illustrated in the $\Delta_1(\mathbf{r})$ and $\Delta_2(\mathbf{r})$ maps of identical regions in Fig. 2b, c.

Zooming out, Fig. 3a maps the combined displacement field $\Delta(\mathbf{r}) = \Delta_1(\mathbf{r}) + \Delta_2(\mathbf{r})$ over a 30 nm field of view, in which a $\Delta_1$-dominant region, readily identified by its transverse polarization relative to $\mathbf{q}_1$ (blue/yellow triangles), occupies the right side of the frame, while a $\Delta_2(\mathbf{r})$-dominant region occupies the upper left corner (red/green triangles). Mapping the displacement magnitudes $|\Delta_1(\mathbf{r})|$ and $|\Delta_2(\mathbf{r})|$ visualizes the striped domain structure, revealing a complex domain morphology with islands of strong modulations (6–11 pm) and basins of PLD suppression (0–3 pm) (Fig. 3b, c). Notably, regions in which both $\Delta_1(\mathbf{r})$ and $\Delta_2(\mathbf{r})$ are present are also observed, such as the bottom left corner of Fig. 3a–c. Quenched disorder tends to broaden phase transitions and favors enhanced isotropy in the nascent-ordered state, and theoretically has been shown to induce apparent fourfold symmetry in 2D striped phases[25–27]. We believe the checkerboard-like regions we observe may result from quenched disorder; varying intensity of atomic columns clearly indicates frozen cation disorder in our data (Supplementary Fig. 17). Alternatively, checkerboard-like ordering could result from projection through stacked $\Delta_1(\mathbf{r})$ and $\Delta_2(\mathbf{r})$ domains in the out-of-plane (b-axis) direction. In either case, the two modulations are predominantly anti-correlated in our data, and we conclude that the symmetry breaking in the disorder-free "clean" limit in this system is very likely striped.

**Nascent order coincident with PLD defects.** CDW domain nucleation near $T_c$ remains a poorly understood process, particularly in the presence of disorder[27,28]. We observe PLD defects coincident with both domain boundaries and nascent domain structures, suggesting their involvement in mediating domain growth and termination. Figure 4 magnifies the region containing a ~5 nm island of $\Delta_2$ order embedded in a $\Delta_1$ domain (Fig. 3, upper white delimiters). Inspection of the $\Delta_1 + \Delta_2$ map (Fig. 4a) reveals shearing in $\Delta_1$ as it passes through the $\Delta_2$ island, evident in the offset of the wavefronts by ~2 atomic rows. Mapping $\Delta_1$

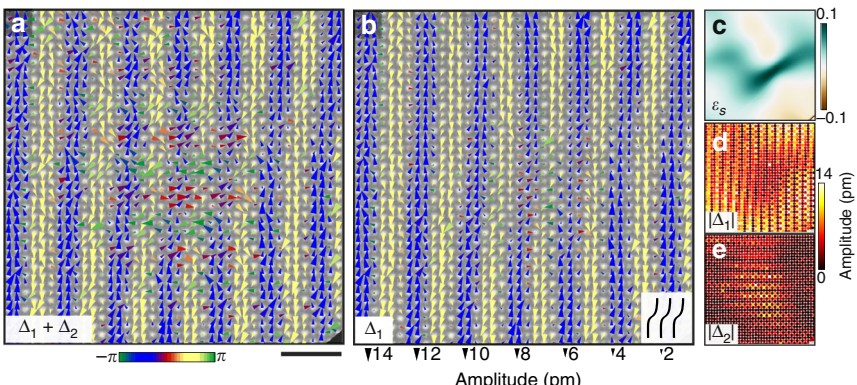

**Fig. 4** Shear deformation coincident with a nascent periodic lattice displacement (PLD) grain. **a** A complete $\Delta = \Delta_1 + \Delta_2$ map of a ~5 nm region of incipient $\Delta_2$ order, and a coinciding shearing of the $\Delta_1$ modulation. **b** A $\Delta_1$ map of the same region highlights the bending wavefronts, and reveals attenuation of the PLD amplitude and some rotation of the displacement vectors in the defective region. **c–e** The shear strain $\varepsilon_s$, $|\Delta_1|$, and $|\Delta_2|$, respectively, in the same region. The maximal shearing aligns with attenuation of $\Delta_1$ and emergence of $\Delta_2$. The scale bar corresponds to 2 nm

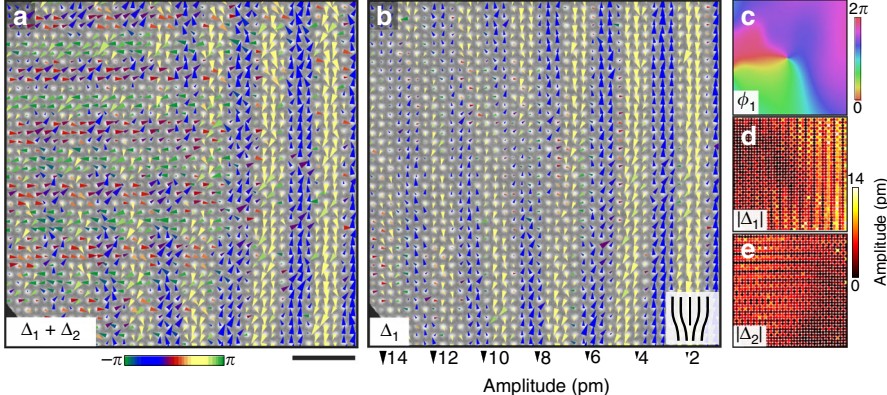

**Fig. 5** Topological singularity coincident with a PLD grain boundary. **a** A complete $\Delta = \Delta_1 + \Delta_2$ map of the interface between a $\Delta_1$-dominant region and coexisting $\Delta_1$ and $\Delta_2$ order. **b** A $\Delta_1$ map of the same region reveals a dislocation in $\Delta_1$, with a Burgers vector of one PLD wavelength, $\lambda_{PLD}\hat{\mathbf{q}}_1$. Analogous to the elastic deformation of an atomic lattice about a crystal dislocation, the elastic response of the PLD includes bending and compression of wavefronts and local displacement rotations. Some attenuation of $\Delta_1$ is apparent in the mixed region. **c–e** The phase $\phi_1$, $|\Delta_1|$, and $|\Delta_2|$, respectively, in the same region. $\Delta_1$ weakens and $\Delta_2$ grows within ~$\lambda_{PLD}$ of the defect core, where $\phi_1$ exhibits an expected $2\pi$ winding. A narrow inlet of $|\Delta_1|$ amplitude collapse extends from the upper left to the singularity. The scale bar corresponds to 2 nm

only (Fig. 4b) accentuates the shear deformation, and exposes $\Delta_1$ attenuation in the strained region, along with rotation of the displacement vectors to roughly align with the local wavefront orientation. To quantify these observations, we map the elastic shear strain field, $\varepsilon_s(\mathbf{r})$, reflecting local bending in the $\Delta_1$ PLD, along with the magnitudes of the two modulations $|\Delta_1|$ and $|\Delta_2|$ (Fig. 4c–e). $\varepsilon_s(\mathbf{r})$ is calculated by extracting the local PLD phase ($\phi \rightarrow \phi(\mathbf{r})$ in Eq. 1)[29] then computing $\varepsilon_s(\mathbf{r}) = \frac{1}{2}\frac{\mathbf{q}_1}{|\mathbf{q}|}\cdot\nabla\phi(\mathbf{r})$ (see Supplementary Note 2)[30,31]. The shear defect plainly coincides with abatement of $\Delta_1$, and strengthening of $\Delta_2$.

Figure 5 magnifies a domain boundary (Fig. 3, lower white delimiters). Exclusive $\Delta_1$ order occupies the right side of the frame in Fig. 5a, while the displacements to the left suggest an intricate interweaving of the two modulations. Mapping $\Delta_1$ only (Fig. 5b) reveals a prominent dislocation in the PLD, in which a single wavefront abruptly terminates. Analogous to edge dislocations in crystalline solids, where the abrupt termination of a row of atoms is accompanied by elastic deformation in the surrounding lattice, we observe elastic deformation of the PLD about the singularity, evident in the warped wavefronts flanking the dislocation core. No defects in the underlying lattice are observed (Supplementary Fig. 17), and the PLD phase $\phi(\mathbf{r})$ exhibits an expected $2\pi$ winding about the discontinuity (Fig. 5c). The interface between the $\Delta_1$-dominant domain and the mixed region occurs within a single PLD wavelength of the defect core, as once again disorder in one modulation accompanies commencement of order in the other. Maps of the PLD magnitudes $|\Delta_1|$ and $|\Delta_2|$ (Fig. 5d, e) reinforce these observations. Moreover, theory predicts modulation amplitude collapse at singularities to prevent divergence of the energy density, and the $|\Delta_1|$ map exhibits a narrow inlet of collapsed amplitude extending from the upper left to the defect core, suggesting complex domain restructuring to accommodate the high-energy feature[30–33]. While displacements at atomic sites directly adjacent to a true singularity will not be accurately reconstructed, we believe the displacements extracted by our method are valid everywhere, because damping and distortion in the defect's central region yields reasonably smooth variations of the displacements (see Supplementary Note 1 and Supplementary Figs. S10, S12, and S13).

## Discussion

In general, many factors appear to govern macroscopic behavior in complex electronic systems. The nanometer scale interplay between new order and defects in an extant order parameter may

be one ubiquitous element, as in emergent charge-ordered states at the core of superconducting vortices, emergent ferromagnetic or superconducting order at CDW discommensuration domain boundaries, or competing PLD domains[6,8,10]. The picture is further complicated by the presence of quenched impurities, which can pin defects, stabilize ordered phases above $T_c$, or lead to complex mixed phases, and may play a role in the phenomena we observe[7,27,32]. Even more fundamental, and still elusive, is a microscopic understanding of which couplings give rise to which competing states, and how. In addition to providing a new structural model of charge-ordered manganites, our data renders the interacting order and disorder in competing PLDs immediately visually apparent: where one modulation bends or "breaks", the other manifests. These observations of the atomically resolved structure of a PLD suggest new lines of inquiry into the nature of modulated phases.

## Methods

**Experimental details.** Bi$_{1−x}$Sr$_{x−y}$Ca$_y$MnO$_3$ (BSCMO) single crystals were grown using the flux method, using Bi$_2$O$_3$, CaCO$_3$, SrCO$_3$, and Mn$_2$O$_3$. Sample preparation for electron microscopy and energy dispersive X-ray spectroscopy (EDX) were performed on a FEI Strata 400 Focused Ion Beam (FIB). From EDX, the composition was determined to be approximately $x = 0.65$ and $y = 0.47$ (Supplementary Fig. 18) with negligible variations over the whole sample (size $0.34 \times 0.28$ mm).

A thin, electron transparent cross section of BSCMO was extracted using FIB lift out, with estimated thickness in the imaging region ranging from 10 to 30 nm. Based on electron diffraction, the orientation of the sample was along the $b$ direction (orthorhombic axis) in the *Pnma* space group (Supplementary Fig. 4). At room temperature (293 K), BSCMO exhibits satellite peaks, indicating the presence of charge ordering.

We performed atomic-resolution imaging in an aberration-corrected scanning transmission electron microscope (FEI Titan Themis) operating at 300 kV. The beam convergence semi-angle was 30 mrad. For Z-contrast imaging, we collected high-angle annular dark-field images, where the inner and outer collection angles were 68 and 340 mrad, respectively. During STEM imaging, the sample experienced an approximately 2 Tesla magnetic field due to its position inside the objective lens, as determined from a Hall bar measurement. In order to minimize the effect of scan noise and stage drift, we acquired 20–30 images in rapid succession with a 2 μs dwell time. We registered and averaged stacks of images using both rigid registration and non-rigid registration methods and found similar results. Data were acquired at 27.4 pm/pixel, and acquisition was optimized for pixel density, field of view, and Fourier space sampling. We performed atom tracking with approximately 2 pm precision (Supplementary Fig. 15 and Supplementary Note 1) by fitting two-dimensional Gaussians to atomic columns using various optimization packages (scipy, photutils, MATLAB) and found consistent results. Atomically resolved EELS spectroscopic mapping was performed in an aberration-corrected NION UltraSTEM at an accelerating voltage of 100 kV and a beam convergence semi-angle of 30 mrad.

**Data availability**. The data that support the findings of this study are available from the corresponding author on reasonable request.

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

## Acknowledgements

We thank Michael J. Zachman and David J. Baek for experimental support. We acknowledge support by the Department of Defense Air Force Office of Scientific Research (FA 9550-16-1-0305) and the Packard Foundation. The FEI Titan Themis 300 was acquired through NSF-MRI-1429155, with additional support from Cornell University, the Weill Institute, and the Kavli Institute at Cornell. This work made use of the Cornell Center for Materials Research Shared Facilities supported through the NSF MRSEC program (DMR-1719875). B.H.S. was supported by NSF GRFP grant DGE-1144153. The work at Rutgers was supported by the Gordon and Betty Moore Foundation's EPiQS Initiative through Grant GBMF4413 to the Rutgers Center for Emergent Materials.

## Author contributions

A.S.A., J.K., and S.-W.C. synthesized the crystals and performed electrical transport characterization. B.H.S., I.E.B., R.H., and L.F.K. acquired and analyzed STEM and electron diffraction data. B.H.S., I.E.B., R.H., and L.F.K. wrote the manuscript. The work was conceived and guided by L.F.K. All authors discussed results and commented on the manuscript.

## Additional information

**Competing interests:** The authors declare no competing financial interests.

