## [Peer Review File · Nature Communications]

Reviewers' Comments:

Reviewer #2:

Remarks to the Author:

Savitzky et al, reported the direct observation of ordered lattice displacement via high resolution STEM and lattice displacement measurements/mapping in $\text{BiO}:35\text{SrO}:18\text{CaO}:47\text{MnO}_3$ system. The authors claimed such strip-like structure observed is related to the charge ordered structure in the system. The careful analysis of the lattice displacement using STEM imaging is commendable. However there are several critical issues that prevent the paper from publication in Nature Comm.

1. The authors spent much time in explaining the method and overall procedure of the lattice displacement measurements in both the text and the supplementary documents. However the key issue here is on how to prove that the observed strip-like structure (periodic lattice displacement structure) is indeed the so-called charge-order structure. Different from other direct imaging tools such as STM, TEM and STEM's existing magnetic field induced by the electromagnetic lenses could have major impact on the charge ordering structure in the materials system in the TEM column. One could imagine that, a Mn containing system, also the system in this study, could lead to major issues in resolving the true nature of the strip structure induced by the charge-ordering in the materials. Thus the reviewer has concerns on the origin of the observed strip-like structure in this study, it might be related to magnetic ordering rather than the charge ordering structures.

2. This work is very narrowly focused on STEM and displacement analysis and weak in all the other physical property measurements and discussions, such as transport properties, magnetic property measurements, specific valence states of the Mn ions in the sample (such as XPS, EELS or others). In addition, is this strip structure uniformly distributed throughout the sample in 3D? If that is the case, XRD reciprocal space mapping could also distinguish such lattice distortion and structural ordering. Those are critical in resolving the underlying mechanisms of the observed strip structure and could strengthen the work significantly. Currently this work solely based on STEM/displacement mapping is thin. Collaboration with a theory team on resolving the mechanisms could also be very beneficial.

3. There are minor editorial related errors in the paper. For example, PLD was cited first before it was fully introduced in the introduction part. There are 4 author affiliation listed and #3 is missing.

Reviewer #3:

Remarks to the Author:

In this paper authors report the direct observation of periodic lattice displacements (PLD) in a charge-ordered manganite via scanning transmission electron microscopy. These are beautiful experiments which allow to visualize shear deformations and topological singularities in the PLD on an atomic scale. While the manganese oxide studied in the paper is a prototype material for charge (and spin) order the issue of CDW's is a hotly debated topic also in the field of high-Tc superconductivity, as mentioned by the authors, and the present paper therefore has the potential of influencing on the field of emergent order far beyond its specific relevance for the manganites. In this context I would also suggest the authors to revise their statement in the abstract that 'charge-ordered phases are a prototypical manifestation of charge-lattice couplings'.

A CDW always manifests via the coupling of charge carriers to the lattice but the latter may not necessarily drive the instability. In general, CDW's may also be generated as a secondary effect of a spin instability (as in one of the 'stripe scenarios' for cuprates) or due to long-range Coulomb interactions which frustrate an underlying phase separation instability. Besides this point I find the paper well written and the original results are presented in a comprehensible form. The supplementary information allows to follow well the analysis of experimental data and eventually reproduce the results.

I therefore recommend publication of the manuscript in Nature Communications. Notice: It seems that affiliation #3 is not specified in the list of addresses.

Reviewer #4:

Remarks to the Author:

The manuscript "Bending and Breaking of Stripes in a Charge-Ordered Manganite" by Benjamin H. Savitzky et al. presents the atomic-scale mapping of periodic lattice displacements (PLDs) in a charge-ordered manganite by aberration corrected scanning transmission electron microscopy. The authors first observed the PLD defects, which provide a clue to the mystery of the competing mechanism in complex materials. The microscopic experiments and corresponding results attract attention from researchers on the exotic electronic systems. The work is interesting and I recommend publication in Nature Communications.

Minor comments:

1. The authors should make mention of the twin boundaries that cannot be avoided in the orthorhombic perovskite oxides. In general, the charge-ordered manganites show the lattice modulation along a-axis (Pnma setting) of the underlying lattice. However, it is not clear whether the image of Fig. 3(a) and the diffraction of Fig. 1(b) exhibit a single underlying lattice or not.
2. In the manuscript, the charge orders are used roughly synonymously with the CDWs. However, the charge-ordered states are physically distinct from the CDW states. These terms should be adequately described.

Reviewer Comments:

Reviewer #2:

Comments:

Savitzky et al, reported the direct observation of ordered lattice displacement via high resolution STEM and lattice displacement measurements/mapping in Bi_{0.35}Sr_{0.18}Ca_{0.47}MnO₃ system. The authors claimed such strip-like structure observed is related to the charge ordered structure in the system. The careful analysis of the lattice displacement using STEM imaging is commendable. However there are several critical issues that prevent the paper from publication in Nature Comm.

1. The authors spent much time in explaining the method and overall procedure of the lattice displacement measurements in both the text and the supplementary documents. However the key issue here is on how to prove that the observed strip-like structure (periodic lattice displacement structure) is indeed the so-called charge-order structure. Different from other direct imaging tools such as STM, TEM and STEMs existing magnetic field induced by the electromagnetic lenses could have major impact on the charge ordering structure in the materials system in the TEM column. One could imagine that, a Mn containing system, also the system in this study, could lead to major issues in resolving the true nature of the strip structure induced by the charge-ordering in the materials. Thus the reviewer has concerns on the origin of the observed strip-like structure in this study, it might be related to magnetic ordering rather than the charge ordering structures.

We agree that clearly establishing the connection between the local structural order observed here and the charge-ordered state is crucial, and thank the reviewer for pointing this out. To this end, we performed two additional experiments to ensure that the ordered structures under study were not induced by the significant magnetic field of the microscope's objective lens. We measured resistivity as a function of temperature both in the absence of a magnetic field, and in the presence of a 2 T field, comparable to that of the objective lens at the sample position, and found the two curves to be nearly equivalent, including the position and functional form of the anomaly associated with the onset of the charge-ordered state. We next performed standard electron diffraction as well as electron diffraction with the objective lens turned off, i.e., in field-free mode, and found the two diffraction patterns to be comparable. In particular, the variations in the superlattice peak intensity as a function of position in momentum space are unchanged, suggesting that the pattern of lattice displacements is unchanged by the applied magnetic field. We believe that the invariability of these two primary experimental signatures of charge-ordering to the 2 T field is very strong indication that PLDs examined in this work are indeed firmly tied to the charge-ordering structure. We further note that the robustness of the charge-ordered state to an applied magnetic field has been previously established in Bi_{1-x}Ca_xMnO₃ and Bi_{0.5}Sr_{0.5}MnO₃, including in fields of up to 45 T (Woo et al, PRB (2001) 63, 134412; Frontera et al, J. Phys.: Condens. Matter (2001) 13, 1071).

These new measurements have been included, with associated discussion, as **Supplementary Figs. S1 and S7**. We have added the following sentences regarding these new transport measurements to the main text: **“Temperature-dependent resistivity and magnetic susceptibility measurements on the host BSCMO crystal reveal an anomaly associated with charge-ordering at T_c = 315 K and 318 K, respectively (Supplemental Figs. S1-S2).**

Transport curves measured at zero field are nearly identical to those measured under application of a 2 T magnetic field, comparable to that of the microscope objective at the position of the specimen (Supplemental Fig. S1).” We have additionally added the following sentence regarding the new, magnetic field-free diffraction experiments to the main text: **“Field-free electron diffraction, with the objective lens turned off, showed no discernible changes in the superlattice structure (Supplemental Fig. S7), consistent with the magnetic field dependent resistivity measurements and suggesting the charge ordered state is robust to the applied magnetic field.”**

Cryogenic STEM measurements provide further confirmation that the PLDs under study are associated with the charge ordered state. By quantitatively characterizing PLD phase disorder at room and cryogenic temperatures, we have found that improvement in local order mirrors sharpening and strengthening of the charge-ordering satellite peaks in diffraction as the temperature is reduced from room temperature (just below T_c) to liquid nitrogen temperatures (well below T_c), painting a structural picture consistent with the broad phase transition suggested by the transport measurements. However, we believe careful analysis of these measurements is beyond the scope of this work, and they are therefore the subject of a separate manuscript and have not been included here.

2. This work is very narrowly focused on STEM and displacement analysis and weak in all the other physical property measurements and discussions, such as transport properties, magnetic property measurements, specific valence states of the Mn ions in the sample (such as XPS, EELS or others). In addition, is this strip structure uniformly distributed throughout the sample in 3D? If that is the case, XRD reciprocal space mapping could also distinguish such lattice distortion and structural ordering. Those are critical in resolving the underlying mechanisms of the observed strip structure and could strengthen the work significantly. Currently this work solely based on STEM/displacement mapping is thin. Collaboration with a theory team on resolving the mechanisms could also be very beneficial.

We performed multiple new measurements in addition to the new transport and diffraction data already discussed. We measured the magnetic susceptibility as a function of temperature, finding an anomaly at a comparable T_c to that found in the resistivity data. This is included as Supplementary Fig. S2. To the main text, we added the sentence: **“Temperature-dependent resistivity and magnetic susceptibility measurements on the host BSCMO crystal reveal an anomaly associated with charge-ordering at $T_c = 315$ K and 318 K, respectively (Supplemental Figs. S1-S2).**” We further performed electron energy loss spectroscopy (EELS) mapping in an aberration corrected NION STEM. Following the work of Loudon et al., which ruled out the possibility of Mn^{3+} - Mn^{4+} ordering and placed an upper bound of ± 0.04 on the degree of possible charge disproportionation in $Bi_{0.5}Sr_{0.4}Ca_{0.1}MnO_3$, we analyzed the new BSCMO EELS data and found no significant periodicity or structure in any valence change in the Mn ions. In the main text, we added the following **“Note that valence modulations have been found to be minimal here (Supplementary Fig. S6) and elsewhere (Loudon et al, PRL (2007) 99, 237205),....”**

We agree that X-ray diffraction is a powerful and important tool to establish the presence of structural modulations in charge-ordered systems and to study the correlation lengths of the modulations. Indeed, various XRD experiments have been performed on $Bi_{1-x}Ca_xMnO_3$ and the isostructural $Re_{1-x}Ca_xMnO_3$ ($Re = La, Nd$) to determine structural ordering associated with charge ordering (Fontera et al, PRB (2001) 64, 5; Radelli et al, PRB (1997) 55, 5; Grenier et al, PRB (2004) 69, 13; Garcia-Munoz et al, PRB (2001) 63, 6; Herrero-Martin et

al, PRB (2004) 70, 2). However, determining the pattern of displacements is extremely challenging, as multiple peaks need to be collected with high resolution (Forgan et al, Nature Communications (2015) 6:10064). Further, nanoscale inhomogeneity and local domain formation significantly complicate structural characterization using spatially-averaged reciprocal space probes (Comin et al, Science (2015) 347, 6228). As shown in the full field-of-view PLD map in Fig. 3 of our submission, the modulation structure is not uniformly distributed in the sample, evidenced by the presence of nanoscale domains and phase defects. We believe our efforts to directly map local modulation structure in real space effectively complement previous reciprocal space based approaches.

We strongly agree with the reviewer that collaboration with a theory team is an important step towards resolving the mechanism underlying the structural modulations we observe here. This is presently ongoing work. Currently, the state of these investigations is preliminary, and we feel that their inclusion herein would be premature.

3. There are minor editorial related errors in the paper. For example, PLD was cited first before it was fully introduced in the introduction part. There are 4 author affiliation listed and #3 is missing.

The definition of PLD has been moved to properly precede its first use, and the author affiliation assignments have been edited to correctly correspond to the author list.

Reviewer #3:

Comments:

In this paper authors report the direct observation of periodic lattice displacements (PLD) in a charge-ordered manganite via scanning transmission electron microscopy. These are beautiful experiments which allow to visualize shear deformations and topological singularities in the PLD on an atomic scale. While the manganese oxide studied in the paper is a prototype material for charge (and spin) order the issue of CDW's is a hotly debated topic also in the field of high-Tc superconductivity, as mentioned by the authors, and the present paper therefore has the potential of influencing on the field of emergent order far beyond its specific relevance for the manganites. In this context I would also suggest the authors to revise their statement in the abstract that "charge-ordered phases are a prototypical manifestation of charge-lattice couplings". A CDW always manifests via the coupling of charge carriers to the lattice but the latter may not necessarily drive the instability. In general, CDWs may also be generated as a secondary effect of a spin instability (as in one of the stripe scenarios for cuprates) or due to long-range Coulomb interactions which frustrate an underlying phase separation instability.

We thank the reviewer for pointing out the relevance of the current work to the broader field of emergent order, and agree that these observations may be of interest to scientists working on many directly and indirectly related systems. Following the reviewer's suggestion, we have edited the abstract to clarify the diverse possible origins of CDW states, as follows: **"In charge-ordered phases, broken translational symmetry emerges from couplings between charge, spin, lattice, or orbital degrees of freedom, giving rise to remarkable phenomena such as colossal magnetoresistance and metal-insulator transitions. The role of the lattice in charge-ordered states remains particularly enigmatic, soliciting characterization of the microscopic lattice behavior."**

Besides this point I find the paper well written and the original results are presented in a comprehensible form. The supplementary information allows to follow well the analysis of experimental data and eventually reproduce the results. I therefore recommend publication of the manuscript in Nature Communications.

Notice: It seems that affiliation #3 is not specified in the list of addresses.

As noted above, the author affiliation list has been edited to correctly correspond to the author list.

Reviewer #4:

Comments:

The manuscript "Bending and Breaking of Stripes in a Charge-Ordered Manganite" by Benjamin H. Savitzky et al. presents the atomic-scale mapping periodic lattice displacements (PLDs) in a charge-ordered manganite by aberration corrected scanning transmission electron microscopy. The authors first observed the PLD defects, which provide a clue to the mystery of the competing mechanism in complex materials. The microscopic experiments and corresponding results attract attention from researchers on the exotic electronic systems. The work is interesting and I recommend publication in Nature Communications.

Minor comments:

1.

The authors should make mention of the twin boundaries that cannot be avoided in the orthorhombic perovskite oxides. In general, the charge-ordered manganites show the lattice modulation along a-axis (Pnma setting) of the underlying lattice. However, it is not clear whether the image of Fig. 3(a) and the diffraction of Fig. 1(b) exhibit a single underlying lattice or not.

All the data presented is from a single domain, however, this was not made clear in the original manuscript. To clarify, we performed linear polarized optical microscopy, which indicates that these crystals contain twin domains with size scales of $\sim 100 \mu\text{m}$. Selected area electron diffraction (SAED) measurements were performed over regions of $1 \mu\text{m}$ diameter, approximately 2 orders of magnitude smaller than the twin domain size, while the regions examined in STEM were $\sim 30 \text{ nm}$ in extent, $\sim 3\text{-}4$ orders of magnitude smaller than the twin domains. The SAED patterns directly exclude any twinning of the a/b or b/c axes in the regions examined. Moreover, varying the regions under study in both diffraction and STEM yield unchanged results, indicating that the effects observed do not result from crystal twinning. Discussion of these points, and the polarized light microscopy results, have been included as Supplementary Fig. S3. To the main text, we have added the sentence:

“Reflective polarized optical microscopy reveals $\sim 100 \mu\text{m}$ twin domains (Supplemental Fig. S3); STEM and electron diffraction are performed within a single twin domain.”

2.

In the manuscript, the charge orders are used roughly synonymously with the CDWs. However, the charge-ordered states are physically distinct from the CDW states. These terms should be adequately described.

We thank the reviewer for pointing out the importance of clarifying our terminology. Clarification is particularly important in the case of CDW states vis-à-vis charge-ordered

states, as a variety of similar but distinct meanings have been attributed to these two terms, both by different authors as well as in different fields, for example in manganite or cuprate investigations. Because our work reports new observations of an emergent local order parameter in a system where the microscopic nature of the ordered state remains contested, we believe broad definitions will best aid ongoing research efforts. Thus, for example, we do not use CDW to refer to order arising from the Fermi nesting phenomenon per se, nor do we use charge-order to necessarily refer to Mn^{3+} - Mn^{4+} ordering, i.e., charge-disproportionation. Rather, we use both terms to equivalently refer to the states associated with empirical observation of satellite peaks indicating some new, emergent order, in concert with an anomaly in the resistivity versus temperature. Physically, this empirical definition broadly includes any generic states in which the translational symmetry is broken by the charge/lattice. Particularly in light of the observations of minimal changes in the Mn valence, we believe choosing model-agnostic terminology is most felicitous here, and we hope that this choice and the corresponding clarifications added to the main text will ensure our work is as useful as possible to other scientists in various related fields.

We have edited the main text in two places to make our terminology as clear as possible. In the introductory section, we have added the following: “Moreover, the contrast mechanism of DF-TEM complicates interpretation, yielding inconsistent models of the modulation structure, **including organization of Mn^{3+} and Mn^{4+} sites, continuous charge density modulations which pin to lattice defects, and Mn^{3+} pairs coupled by an adjoining hole^{13,14,15,17}.**” In the beginning of the results section, we have added the sentence: “**Note that valence modulations have been found to be minimal here (Supplementary Fig. S6) and elsewhere¹⁵, therefore the state giving rise to the observed satellite peaks and accompanying the resistivity anomaly is referred to empirically as the charge-ordered or CDW state, agnostic to a particular underlying model.**”

Reviewers' Comments:

Reviewer #2:

Remarks to the Author:

The authors have made effort in addressing the comments and now the paper is in a better form for publication.

Reviewer #3:

Remarks to the Author:

Authors have included the suggestions of my previous report in the manuscript and have, in my opinion, also convincingly addressed the issues raised by the other two referees, both in the manuscript and supplementary material. I therefore recommend publication of the manuscript in Nature Communications.

Reviewer #4:

Remarks to the Author:

The manuscript has been much improved and is in a nice condition now.